# Yeast-Hydrolysate-Derived 1-Methyl-1,2,3,4-tetrahydro-β-carboline-3-carboxylic Acid Inhibits Fat Accumulation during Adipocyte Differentiation

**DOI:** 10.3390/foods12183466

**Published:** 2023-09-18

**Authors:** Nari Kim, Sekyung Lee, Eun-Jin Jung, Eun Young Jung, Un-Jae Chang, Cheng-Min Jin, Hyung Joo Suh, Hyeon-Son Choi

**Affiliations:** 1Department of Integrated Biomedical and Life Science, Graduate School, Korea University, Seoul 02841, Republic of Korea; 84hurizia@cremar.co.kr (N.K.); appjn@naver.com (S.L.); 2Transdisciplinary Major in Learning Health Systems, Department of Healthcare Sciences, Graduate School, Korea University, Seoul 02841, Republic of Korea; 3Department of Food and Biotechnology, Korea University, Sejong 30019, Republic of Korea; ejjung1124@korea.ac.kr; 4Department of Home Economic Education, Jeonju University, Jeonju 55069, Republic of Korea; jjjj@jj.ac.kr; 5Department of Food and Nutrition, Dongduk Women’s University, Seoul 02748, Republic of Korea; uj@dongduk.ac.kr; 6Analysis and Research Department, NeuroVIS, Inc., Hwaseong-si 18469, Republic of Korea; smkim@neurovis.kr; 7Department of Food Nutrition, Sangmyung University, Hongjimun 2-Gil 20, Jongno-Gu, Seoul 03016, Republic of Korea

**Keywords:** yeast hydrolysate (YH), adipogenic factor, lipid synthesis, SREBPs, MTCA

## Abstract

This study aimed to investigate the impact of yeast hydrolysate (YH) on lipogenesis, elucidate its mechanistic action, and identify the active compounds responsible for its anti-adipogenic effects. YH (2 mg/mL) significantly reduced Oil Red O-stained lipids. YH (2 mg/mL) also downregulated C/EBPβ and upregulated KLF2, both of which are early adipogenic factors. Moreover, YH (2 mg/mL) decreased C/EBPα, PPARγ, FABP4, FAS, ACC, and HMGCR mRNA expression. Additionally, YH significantly downregulated SEBP1c and SREBP2 and their target genes, which govern fatty acid and cholesterol metabolism; however, 2 mg/mL YH had a greater suppressive effect on SREBP1c than on SREBP2. YH (2 mg/mL) also significantly reduced the mRNA level of G6PD and malic enzyme, which are enzymes that synthesize NADPH for lipid synthesis, compared with the control. Furthermore, 1-methyl-1,2,3,4-tetrahydro-β-carboline-3-carboxylic acid (MTCA) was identified as the active compound with anti-adipogenic effects using solvent fractionation and chromatographic analysis of YH, and 1.1 μg/mL MTCA significantly downregulated SREBP1c/SREBP2 mRNAs by 47.8% and 69.2%, respectively, along with the target genes FAS, ACC, and HMGCR by 79.0%, 77.0%, and 40.9%, respectively. Collectively, YH effectively suppressed adipogenic lipid storage by downregulating SREBP- and NADPH-synthesizing genes. These findings suggest that YH containing MTCA has the potential to act as an anti-obesity agent.

## 1. Introduction

Obesity is a disease involving excessive lipid accumulation in the body owing to an imbalance between energy input and consumption [1,2]. It is caused by various factors, such as genetics, dietary habits, and environmental influences [1,2]. The incidence of obesity is steadily increasing, and it is closely associated with hyperlipidemia, cardiovascular disease, hypertension, diabetes, fatty liver, and arteriosclerosis [3,4].

Global estimates from the World Health Organization (WHO) highlight the widespread problem of obesity. In 2016, the data showed that 1.9 billion adults aged 18 and older were overweight (BMI ≥ 25), with over 650 million adults classified as obese (BMI ≥ 30). This means that, on a global scale, roughly 11% of adult men and 15% of adult women were categorized as obese [5]. Based on the 2018 noncommunicable disease country profiles published by the WHO, the occurrence of obesity displayed significant variation among different regions and countries. The prevalence of overweight or obese individuals exceeded 70% in Mexico (74.1%) and the United States (73.1%), while both Japan (27.2%) and South Korea (37.8%) remained below the 38% mark. However, obesity is steadily on the rise [5].

Excessive fat storage during obesity is characterized by adipocyte hypertrophy and hyperplasia, with hypertrophy being more common than hyperplasia after adulthood [6]. Lipid accumulation is associated with the expression of various genes involved in lipid synthesis during adipocyte differentiation. The expression of adipogenic factors, such as PPARγ and C/EBPα, increases to induce lipid storage [7,8]. Lipid synthesis is regulated by two master genes, SREBP1 and SREBP2. SREBP1 is a transcription factor that regulates fatty acid synthesis, while SREBP2 is a nuclear factor that regulates cholesterol synthesis [9].

Anti-obesity drugs targeting fat-synthesis-related factors have been developed, but they have various side effects, such as dizziness, headache, steatorrhea, and depression, which may offset their effectiveness [10]. Therefore, bioactive dietary materials that control obesity and obesity-related metabolic diseases and have fewer side effects have received considerable attention [11]. Food-based peptides are used in the fields of nutrition, food science, and health [10,11]. Active peptides exhibit a variety of physiological activities, such as antidiabetic, antihypertensive, and anticancer activities, with desirable bioavailability [12,13]. The molecular weight of an active peptide is generally less than 10 kDa [14]. 

β-Carboline alkaloids are naturally occurring tryptophan derivatives found in various plants and microorganisms [15,16]. Alkaloids are derived from the proteolytic products of proteins and amino acids with heterocyclic amines [17]. Alkaloids have attracted considerable attention because of their physiological activities, including anxiolytic, anticonvulsant, antitumor, and antimicrobial functions [18]. Recently, garlic-containing β-carboline was found to exert anti-adipogenic effects [19]. On the other hand, β-Cabolines derived from carbohydrates have been found to form through the reaction of tryptophan and 3-deoxyglucosone. These b-carbolines have been recognized as a type of advanced glycation end product (AGE) [20]. AGEs are associated with the development of various diseases, including diabetes, cardiovascular illness, and obesity [20]. However, there is a significant controversy surrounding the relationship between AGEs and obesity. While some studies have reported that a higher intake of AGEs can lead to complications related to obesity [21], a recent meta-analysis has shown an inverse correlation between circulating AGEs and body mass index (BMI), indicating that AGEs could have the potential to cause obesity [22].

*Saccharomyces cerevisiae*, a type of yeast, has long been used as an experimental model in the field of molecular biology and manufacturing of food, such as liquor and bread [13]. Yeast contains approximately 50% proteins, lipids, RNA, nucleic acids, vitamins, minerals, and enzymes; therefore, it has a high utilization value as a functional raw material and is recognized as a GRAS by the US FDA [23]. Yeast hydrolysates (YHs) reportedly have various physiological activities, such as radical scavenging activity and immune stimulating effect [13]. YHs of less than 10 kDa have been shown to inhibit fat accumulation in obese rats and humans [24,25,26]. While YH has been studied for its anti-obesity effects at both the preclinical and clinical levels, there remains a gap in understanding its cellular-level inhibition of obesity and the associated mechanisms. Furthermore, there have been no studies exploring the active compounds derived from yeast hydrolysates. Therefore, the primary objective of this study is to identify active compounds within YH that inhibit lipid accumulation and elucidate the underlying mechanisms.

## 2. Materials and Methods

### 2.1. Preparation of Yeast Hydrolysate

Yeast hydrolysate was prepared with slight modifications to the previous method. *Saccharomyces cerevisiae* cultured by Choheung Chemical was treated with proteolytic enzymes, and yeast hydrolysates of 10 kDa or less were recovered using a 10 kDa molecular weight cut-off membrane (Nanofiltration System, SEC, Republic of Korea) and then spray dried. The crude protein content of YH was 550 mg/g and the active substance, MTCA, was 0.52 mg/g.

### 2.2. Cell Culture

3T3-L1 cells were purchased from Korea Cell Line Bank (Seoul, Republic of Korea) and cultured in DMEM containing 10% FBS (Gibco BRL, Grand Island, NY, USA) at 5% CO_2_ and 37 °C. The cells were cultured in a 12-well plate and allowed to reach 100% confluence, maintaining for 48 h. The medium was changed with DMEM containing the differentiation inducers IBMX (0.5 mM; Sigma-Aldrich, St. Louis, MO, USA), dexamethasone (0.25 μM; Sigma-Aldrich), and insulin (5 μg/mL; Sigma-Aldrich) with or without YH (NeoCrema Co., Ltd., Seoul, Republic of Korea). After 72 h, the medium was supplemented with insulin alone every two days to promote differentiation. Differentiation was performed for eight days in total. 

### 2.3. WST-1 Assay

3T3-L1 cells (3.8 × 10^5^ cells/mL) were seeded in a 12-well plate and cultured for 24 h. Then, the cells were treated with YH (1.0–4.0 mg/mL) followed by 10% (*v/v*) WST-1 solution and a 60 min incubation at 5% CO_2_ and 37 °C. Each sample (100 μL) was moved to a 96-well plate. The absorbance at 450 nm was measured using a microplate reader (Tecan, Grödig, Austria). 

### 2.4. Oil Red O (ORO) Staining

To evaluate the inhibitory effect of YH on lipid storage in 3T3-L1 cells, ORO staining was performed as described previously [27]. After inducing cell differentiation, the medium was removed from the 6-well plate and the cells were washed with PBS (HyClone Laboratories Inc., Logan, UT, USA). Subsequently, the cells were fixed with 1 mL of 10% formalin solution (Tech & Innovation, Chuncheon, Republic of Korea) at 25 °C for 10 min. The formalin solution was then replaced with 1 mL of ORO solution in 2-propanol (0.5%, *v*/*v*) and incubated for 1 h. The solution was then removed, and the plate was dried. ORO was eluted using 2-propanol and the absorbance at 540 nm was measured using a microplate reader (Tecan, Männedorf, Swizerland).

### 2.5. Real-Time PCR

Cells differentiated in the presence or absence of YH were washed with PBS and harvested by scrapping. Thereafter, RNA was extracted using a Total RNA Extraction Kit (Intron) and 1 μg RNA was used to synthesize DNA using a cDNA kit (Thermo Fisher Scientific, Waltham, MA, USA). The synthesized DNA was amplified using each primer (Table 1), DEPC water, a one-step SYBR PrimeScriptTM RT-PCR Kit (Takara, Dalian, China), and a Real-Time PCR Detection System (Bio-Rad). The PCR reaction was performed under the following conditions: 42 °C for 5 min, 95 °C for 10 s, and 35 cycles of 95 °C for 5 s and 60 °C for 20 s. The data were analyzed using the 2^−(∆∆CT)^ method [28], and the relative mRNA expression was compared to that of the control group.

### 2.6. Western Blot Analysis

Proteins (50 μg) were separated using SDS-PAGE and transferred onto a nitrocellulose membrane (Hercules, CA, USA). The membrane was subsequently incubated with 5% skim milk for 1 h and washed thrice with 1X Tris-buffered saline with tween-20 buffer (TBS-T) for 10 min. The membrane was incubated with primary-antibody-containing milk solution overnight at 4 °C, washed thrice with TBS-T, and reacted with secondary antibody for 1 h at 25 °C. ECL solution (Thermo Fisher Scientific) was added to the membrane to activate horseradish peroxidase bound to the secondary antibody. Protein signals were visualized using the LAS imaging system (Fuji, New York, NY, USA) and quantified using ImageJ (Bethesda, MD, USA, National Institute of Health).

### 2.7. Identification of Active Compounds 

YH (25 g) was sequentially fractionated using solvents of different polarities. In particular, YH (25 g) was subjected to solvent fractionation using ethyl acetate (3 × 800 mL) and butanol (3 × 800 mL), and each fraction was concentrated and lyophilized. For the butanol fraction, the active compound was purified using reverse-phase (RP)-C18 silica gel (Merck, 230–400 mesh, 10% MeOH → 100% MeOH) and Sephadex LH-20 (Pharmacia, Uppsala, Sweden, 10% MeOH isocratic system, 100% MeOH for washing). The fraction obtained using the Sephadex LH-20 process was purified using an Agilent Eclipse C18 column (250 × 21.2 mm) and semi-preparative HPLC (SPD-20A/20AV Series Prominence HPLC UV-Vis Detectors; Shimadzu, Tokyo, Japan). Each fraction was confirmed using LC–MS and thin-layer chromatography (TLC). LC–MS was run using an analytical Kinetex^®^ 5 µm C18100 Å column (5 µm, 2.1 × 100 mm, Phenomenex, Torrance, CA, USA), while TLC was performed using silica gel F254 and RP-18 F254s plates from Merck. The results were detected either under UV light or by heating after spraying with anisaldehyde–sulfuric acid.

### 2.8. Statistical Analysis

All experimental results were analyzed using SPSS ver. 18.0 (SPSS Inc., Chicago, IL, USA) and are displayed as the mean ± standard deviation (SD). The statistical significance of the data in each experimental group was verified at *p* < 0.05 and analyzed using analysis of variance (ANOVA) followed by Tukey’s multiple range test.

## 3. Results

### 3.1. Effect of YH on Fat Storage

To examine the viability of 3T3-L1 cells, they were treated with different concentrations of YH (0–4 mg/mL) for 72 h. YH decreased the cell survival rate in a concentration-dependent manner. The cell viability following 1 and 2 mg/mL YH treatment was 92.2% and 90.2%, respectively, which is not significantly different compared to that of the control (100%; Figure 1). On the contrary, YH at the concentration of 3 mg/mL or more rapidly decreased the cell survival rate. Therefore, subsequent experiments were performed using 1 and 2 mg/mL YH. YH effectively reduced lipid accumulation during adipogenesis (Figure 1; *p* < 0.001, respectively). In particular, 1 and 2 mg/mL YH reduced ORO-stained fat by 28.5% and 35.7%, respectively (Figure 1). This result showed that YH effectively suppresses the lipid synthesis during differentiation without a significant change of cell viability in adipocyte. 

### 3.2. Effect of YH on the mRNA Expression of Adipogenic Factors 

YH significantly decreased the mRNA expression of adipogenic factors. Both concentrations of YH (1 and 2 mg/mL) reduced the expression of C/EBPβ, an early adipogenic factor, by 55.6% and 90.4%, respectively, compared to the control group (Figure 2). Contrarily, 2 mg/mL YH increased the mRNA expression of KLF2, an early anti-adipogenic factor, by 160.4% (Figure 2). Furthermore, YH significantly reduced the mRNA level of the late differentiation factors C/EBPα and PPARγ and their target FABP4 (Figure 2). In particular, 1 and 2 mg/mL YH significantly decreased the mRNA expression of C/EBPα by 39.0% and 89.0%, respectively, and PPARγ by 87.8% and 88.4%, respectively, compared to the control group. Additionally, FABP4 mRNA expression was significantly decreased by 2 mg/mL YH by 33.8% compared to that of the control group. These results show that YH inhibits lipid accumulation in adipocytes by regulating adipogenic factors involved in both early and late differentiation stages.

### 3.3. Effect of YH on the Expression of Fatty Acid/Cholesterol Synthetic Genes

YH significantly decreased the mRNA expression of FAS and ACC, the main enzymes involved in fatty acid synthesis. In particular, 1 and 2 mg/mL YH reduced the mRNA expression of FAS by 73.8% and 92.9%, respectively, and ACC by 25.9% and 74.8%, respectively, compared to the control group (Figure 3). Additionally, YH downregulated SREBP1c, a transcription factor that regulates fatty acid synthesis, at both the mRNA (Figure 3) and protein (Figure 4) levels. YH at concentrations of 1 and 2 mg/mL decreased SREBP1c mRNA expression by 55.4% and 79.9%, respectively, and protein abundance by 38.0% and 40.0%, respectively. These results suggest that YH inhibits fatty acid synthesis by regulating SREBP1. Furthermore, YH significantly downregulated SREBP2, a transcription factor involved in cholesterol synthesis, at both the mRNA and protein levels (Figure 3 and Figure 4). A higher concentration of YH (2 mg/mL) reduced SREBP2 mRNA and protein expression by 56.8% and 27.1%, respectively, compared to the control group. YH-mediated suppression of SREBP2 reduced the mRNA expression of HMGCR, a key enzyme in cholesterol synthesis. In particular, 2 mg/mL YH reduced HMGCR mRNA expression by 81.8%. These results show that YH suppresses fat synthesis by regulating SREBP1 and SREBP2 at the mRNA and protein levels.

### 3.4. Effect of YH on the Expression of Lipogenic Cofactors 

YH suppressed the mRNA expression of G6PD and ME, which are enzymes involved in the production of NADPH, a cofactor required for fatty acid and cholesterol synthesis (Figure 5). YH (1 and 2 mg/mL) decreased the mRNA expression of G6PD by 38.4% and 53.8%, respectively, and ME by 78.4% and 91.4%, respectively, compared to the control group. YH not only regulated the expression of lipid synthetic genes, but also regulated the expression of G6PD and ME. This result indicates that YH-mediated inhibition of G-6PD and ME may contribute to the suppression of lipid accumulation through the NADPH decrease. 

### 3.5. Isolation and Identification of the Active Compound in YH

YH was subjected to solvent fractionation, and the butanol fraction (19.7 g) was obtained through successive treatment with ethyl acetate and butanol. Next, 9.4 g of the butanol fraction was further subjected to RP-C18 silica gel chromatography, resulting in a separation into eight fractions (Appendix A). Among them, BR-7 exhibited a significant inhibitory effect on fat accumulation in 3T3L1 cells during adipocyte differentiation as evidenced by ORO staining. BR-7 was further fractionated using Sephadex LH-20 chromatography to obtain four fractions (BR7-1 to BR7-4), and among them, the BR7-2 fraction showed a high fat accumulation inhibitory effect. Among the five fractions of BR7-2 (BR7-2a–BR7-2e) obtained using Preparative HPLC, BR7-2b showed the highest fat accumulation inhibitory effect. BR7-2b was further fractionated (BR7-2b1, BR7-2b2, and BR7-2b3) using semi-preparative HPLC (Figure 6). Among the three isolated fractions, BR7-2b1 and BR7-2b2 showed an inhibitory effect on fat accumulation. LC–MS (Figure 6) and 1H-NMR (Appendix A) revealed that BR7-2b1 and BR7-2b2 were (1S,3S)-1-methyl-1,2,3,4-tetrahydro-β-carboline-3-carboxylic acid (MTCA) and (1R,3S)-1-methyl-1,2,3,4-tetrahydro-β-carboline-3-carboxylic acid, respectively.

### 3.6. Effect of MTCA on the Expression of Lipid Synthetic Genes

The expression of lipid-synthesis-related genes in adipocytes was analyzed using 0.55 and 1.10 μg/mL MTCA, which correspond to 1 and 2 mg/mL YH, respectively. The expression of SREBP1c, FAS, and ACC, which are in charge of fatty acid synthesis, was significantly reduced (Figure 7). In particular, 1.1 μg/mL MTCA decreased SREBP1c, FAS, and ACC mRNA expression by 47.8%, 79.0%, and 77.0%, respectively, compared to the control. Similarly, the expression of the cholesterol-synthesis-related genes SREBP2 and HMGCR was significantly decreased (Figure 7). Treatment with 1.10 µg/mL MTCA showed 69.2% and 40.9% inhibitory effects on SREBP2 and HMGCR mRNA expression, respectively. These data suggest that YH-mediated lipid-suppressive effects are attributed to MTCA, the active compound in YH.

## 4. Discussion

YH has long been used as a food material and is known to have anti-obesity activity [24,25,26]. In obese ICR mice, YH administration reduced serum triglyceride levels and body weight by inhibiting hepatic G6PD and ME [25]. These results suggest that the downregulation of G6PD and ME in the liver tissue may suppress the production of NADPH, which is essential for fat synthesis, thereby contributing to weight loss. This observation is consistent with the current study findings. Furthermore, in a clinical study, YH administration at 1.0 g/day for 10 weeks reduced the weight of obese adults by 2.60 kg along with accumulated abdominal fat [24]. Anti-obesity effects of YH have been verified in animals and humans, but the inhibition of G6PD and ME has not. Moreover, studies on the mechanisms of fat reduction are lacking. 

In this study, we analyzed the mRNA expression of factors involved in fat synthesis and found that YH regulated the expression of genes responsible for fatty acid and cholesterol syntheses. Lipid droplets formed during adipocyte differentiation are cellular organelles composed of a monolayer of phospholipids and neutral lipids, including triglycerides and cholesterol esters [28,29]. Enzymes, such as diglyceride acyltransferase 1, induce the formation of lipids in the endoplasmic reticulum [29,30]. The lipid droplet storage function is most prominent in white adipocytes, which are characterized by abnormal lipid accumulation and directly associated with obesity [31]. 

Thus, adipocyte differentiation has emerged as a focal point in the quest to discover materials to control obesity and its associated diseases. Numerous plant-based materials have been investigated for their potential to inhibit adipocyte differentiation. A recent study showed the anti-adipogenic effect of night-blooming jasmine [32]. In addition, lipid droplets have an endocrine function, in that they release proteins, such as adipokines and cytokines, that affect metabolic regulation [33]. Leptin is a representative adipokine derived from adipocyte. It plays a critical role in regulating appetite and energy balance. When leptin levels are elevated, the brain perceives the body as having sufficient energy reserves, subsequently reducing appetite and increasing energy expenditure to maintain body weight. However, leptin resistance is commonly associated with obesity. Despite the presence of high level of leptin, the brain does not respond properly to the signal. This results in impaired appetite and energy balance control, leading to overeating and weight gain [34].

Various hormones and cell proliferation inducers induce adipocyte differentiation, which is associated with excessive lipid accumulation in the adipose tissue [35]. Adipocyte differentiation is accompanied by a series of transcriptional actions involving various adipogenic factors, such as C/EBPα and PPARγ. The expression of early differentiation factors (C/EBPβ and C/EBPδ) increases during 3T3L1 cell differentiation, subsequently regulating late adipogenic factors (C/EBPα and PPARγ) [35]. C/EBPα and PPARγ exert synergistic effects on each other while interacting during transcription. Consequently, various enzymes involved in fat transport, synthesis, and globule formation are induced [36,37]. C/EBPα and PPARγ are regulated by pro-early adipogenic regulators, such as Krox20, KLF 5, and KLF15, and anti-early adipogenic regulators, such as CHOP and KLF2, which are upstream in the adipogenesis pathway [38]. In the present study, YH inhibited adipogenesis by increasing the expression of the inhibitory regulator KLF2 and suppressing the expression of C/EBPβ, a positive differentiation regulator. Moreover, YH controlled adipogenesis by regulating the initial stages of adipogenesis. 

Furthermore, YH inhibited gene and protein expression of SREBP1c and SREBP2, which are responsible for fatty acid and cholesterol syntheses, respectively. SREBP-1c plays a critical role in fatty acid metabolism and lipogenesis, whereas SREBP-2 primarily regulates genes associated with cholesterol homeostasis [39]. SREBP1c is a nuclear factor that promotes the mRNA expression of FAS and ACC, enzymes that promote the synthesis of fatty acids, which are the major components of lipids, thereby facilitating triglyceride synthesis. SREBP-1c is mainly expressed at the early stage of adipocyte 3T3L1 differentiation and stimulates the induction of C/EBPα and PPARγ, which is promoted by C/EBPβ and C/EBPδ, initiating the adipogenic process [40,41]. SREBP-1c is abundant in the adipose tissue and liver and regulates lipid metabolism by increasing its activity and expression in response to insulin [42,43].

SREBP2 governs cholesterol metabolism by promoting HMGCR gene expression [44,45]. SREBP2 responds to intracellular cholesterol levels and shuttles between the endoplasmic reticulum and nucleus based on cellular requirements. In the nucleus, it interacts with the sterol response element located in the promoter region of target genes, such as HMGCR, and promotes gene expression [46]. In the present study, YH effectively decreased the gene and protein expression of SREBP1c and SREBP2 to downregulate FAS and HMGCR, the main enzymes involved in fatty acid and cholesterol syntheses. Therefore, YH-mediated regulation of SREBPs and their target genes may contribute to the inhibitory effect of YH on intracellular fat accumulation.

Saccharomyces cerevisiae has been mainly used to produce yeast extract or hydrolysates, but different yeasts such as Kluyveromyces and Candida have also been reported as sources to prepare yeast hydrolysates. The fermentation with Candida intermedia LAMAP1790 improved the antioxidant and angiotensin-converting enzyme inhibitory effects compared to the fermented control of kefir grain [47]. The peptides from Chhupi, a traditional cheese in Bhutan and Nepal, which was fermented with Kluyveromyces marxianus YMP45 and Saccharomyces cerevisiae, have demonstrated DPPH and Superoxide radical scavenging activities with IC50 values ranging from 0.38 to 0.75 mg protein and 0.29 to 0.32 mg protein, respectively [48].

In this study, MTCA was isolated from YH and identified as an active compound that caused the inhibition of fat accumulation. The effect of MTCA was confirmed by analyzing the expression of genes encoding adipogenic factors and those related to lipid synthesis. MTCA is abundant in various food sources, including soy sauce, alcoholic beverages, and citrus fruits [15,18,49]. MTCA is a tryptophan derivative also commonly found in fermentation products of *S. cerevisiae* [16,50]. MTCA is synthesized from tryptophan and acetaldehyde via the Pictet–Spengler reaction in baker’s yeast [46]. MTCA is a pivotal compound in the advancement of new materials because of its diverse biological and pharmacological activities [15]. 

A recent study reported the isolation and identification of MTCA from garlic and demonstrated its anti-adipogenic effect, which is consistent with current study findings [19]. However, these two studies exhibit several differences. First, garlic-derived MTCA reduced the levels of adipogenic and lipolytic genes, including C/EBPα, HSL, and ATGL, at concentrations ranging from 10 to 40 μM. Contrarily, YH-derived MTCA showed an anti-adipogenic effect at concentrations as low as 2–5 μM, downregulating adipogenic and lipogenic factors. Second, the MTCA content in garlic and YH may be different. The yield of YH-derived MTCA was 0.2% using solvent fractionation and chromatography, whereas the yield of garlic-derived MTCA has not been reported. Citrus fruits and juices are rich sources of MTCA, with grapefruit containing 3.5 μg/g and orange juice containing 2.9 mL/L MTCA [16]. Further studies should analyze MTCA content in various dietary sources, including garlic. 

There are some limitations in this study. First, most of the analysis in this study was conducted at the mRNA levels. While gene expression is crucial for understanding adipogenic regulation, it is essential to confirm YH- or MTCA-mediated regulation at the protein or active molecule level, as these are functional molecules. Second, YH was shown to regulate early adipogenic factors such as C/EBPβ andKLF2, indicating that early adipogenic steps are the target of YH to suppress lipid accumulation. To support this finding, it is advisable to examine each stage of adipogenesis using Oil red O (ORO) staining to provide a more comprehensive view. Although YH was observed to regulate various factors involved in the adipocyte differentiation and lipid synthesis, there is a lack of analysis regarding the primary mechanisms responsible for its anti-adipogenic effect. It is crucial to conduct a detailed analysis of signaling pathways involved in the adipogenic process or lipid synthesis. Furthermore, additional studies should be performed using MTCA isolated as an active compound for its anti-adipogenic effect. Specifically, investigations involving animal and clinical studies are necessary to confirm the anti-obesity effect of MTCA.

In conclusion, YH exerted anti-adipogenic effects by downregulating adipogenic and lipogenic genes, namely SREBP1c and SREBP2. YH-mediated anti-adipogenic effects may be attributed to MTCA, the active compound isolated from YH. Therefore, YH-containing MCTA has the potential to be an edible agent for controlling obesity and metabolic diseases.

## Figures and Tables

**Figure 1 foods-12-03466-f001:**
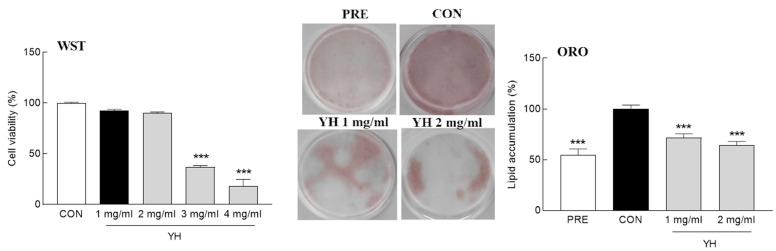
Effect of yeast hydrolysate (YH) on cell viability and lipid accumulation in 3T3L1 cells. CON: no YH treatment; PRE: 3T3L1 preadipocyte. Data are presented as the mean ± standard deviation (SD). *** *p* < 0.001 vs. CON (Tukey’s multiple range test).

**Figure 2 foods-12-03466-f002:**
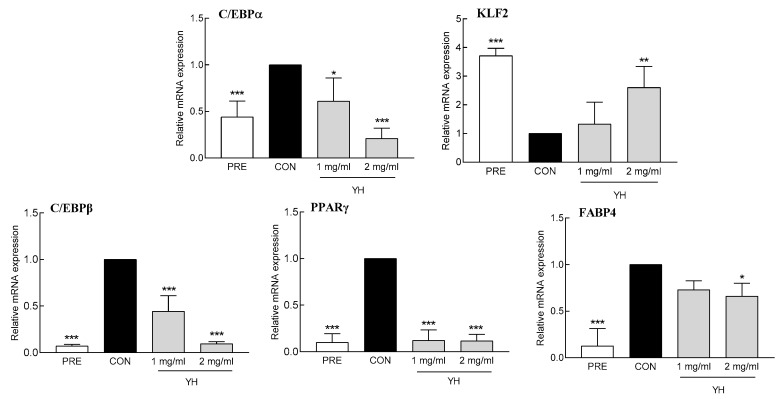
Effect of YH on 3T3L1 cell differentiation factors. PRE: 3T3L1 preadipocyte, CON: no YH treatment. Data are presented as the mean ± SD. * *p* < 0.05, ** *p* < 0.01, and *** *p* < 0.001 vs. CON (Tukey’s multiple range test).

**Figure 3 foods-12-03466-f003:**
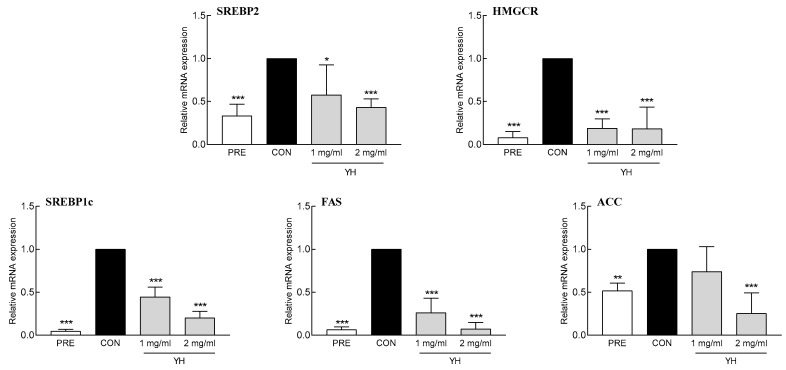
Effect of YH on factors related to lipid synthesis (SREBP2, HMGCR, SREBP1c, FAS, and ACC) in 3T3L1 cells. PRE: 3T3L1 preadipocyte, CON: no YH treatment. Data are presented as the mean ± SD. * *p* < 0.05, ** *p* < 0.01, and *** *p* < 0.001 vs. CON (Tukey’s multiple range test).

**Figure 4 foods-12-03466-f004:**
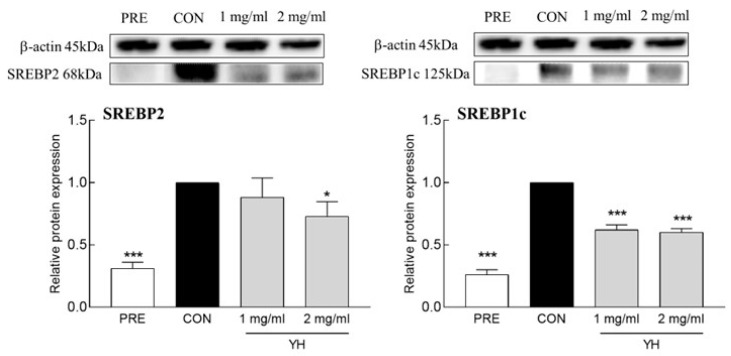
Effect of YH on the protein expression of SREBP1c and SREBP2. CON: no YH treatment. Data are presented as the mean ± SD. * *p* < 0.05 and *** *p* < 0.001 vs. CON (Tukey’s multiple range test).

**Figure 5 foods-12-03466-f005:**
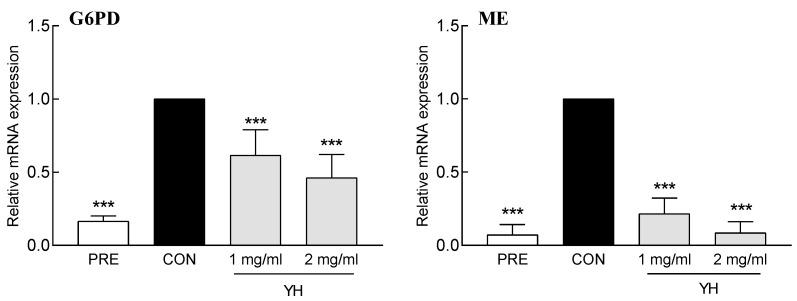
Effect of YH on glucose-6-phosphate dehydrogenase (G6PD) and malic enzyme (ME) expression in 3T3L1 cells. PRE: 3T3L1 preadipocyte, CON: no YH treatment. Data are presented as the mean ± SD. *** *p* < 0.001 vs. CON (Tukey’s multiple range test).

**Figure 6 foods-12-03466-f006:**
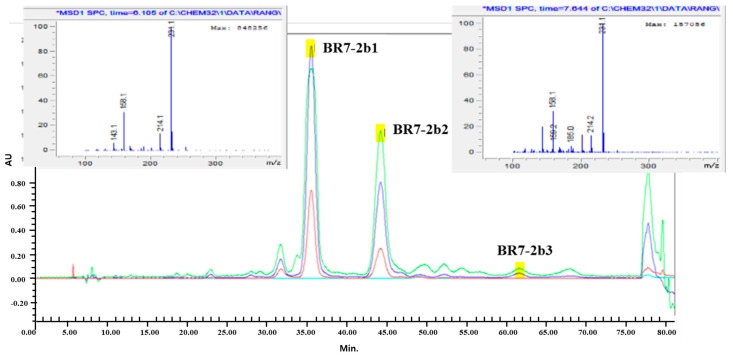
Chromatogram of the active compound fraction BR7-2b using semi-preparative HPLC.

**Figure 7 foods-12-03466-f007:**
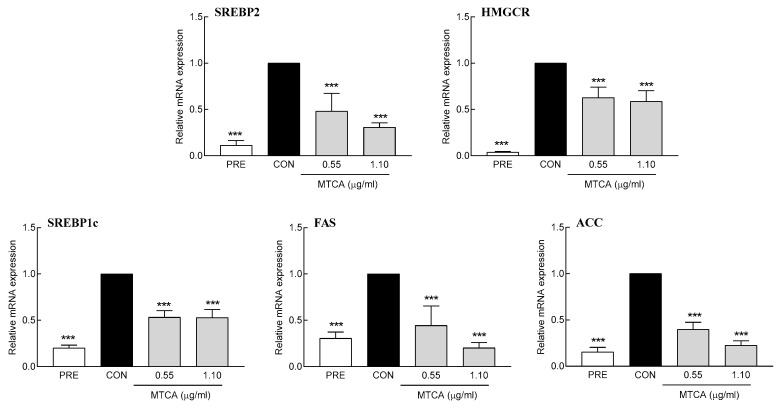
Effect of 1-methyl-1,2,3,4-tetrahydro-β-carboline-3-carboxylic acid (MTCA) on lipogenic factors. PRE: 3T3L1 preadipocyte, CON: no YH treatment. Data are presented as the mean ± SD. *** *p* < 0.001 vs. CON (Tukey’s multiple range test).

**Table 1 foods-12-03466-t001:** Primer list.

Gene	Primer	Primer Sequence (5’-3’)
GAPDH	F	CATCACTGCCACCCAGAAGACTG
	R	ATGCCAGTGAGCTTCCCGTTCAG
FAS	F	TGCTTGCTGGCTCACAGTTA
	R	ATCAGTTTCACGAACCCGCC
G6PD	F	GACCAAGAAGCCTGGCATGTTC
	R	AGACATCCAGGATGAGGCGTTC
SREBP2	F	TGTCGCACTGCAAAGGGAG
	R	GCTCCCTAGTCTGTACCCGA
HMGCR	F	GCTCGTCTACAGAAACTCCACG
	R	GCTTCAGCAGTGCTTTCTCCGT
SREBP1-C	F	GGGGAACTTTTCCTTAACGTGG
	R	TCCAGTTCGCACATCTCGG
ME	F	AGAGCAGTGCTACAAGGTGACC
	R	CCAAGAGCAACTCCAGGGAACA
c/EBPβ	F	CAAGATGCGCAACCTGGAGA
	R	GACAGCTGCTCCACCTTCTT
KLF2	F	CTCTCCATGGGATTGGACGG
	R	TCCGGGTAGTAGAAGGCAGG
PPARΥ	F	TTCGATCCGTAGAAGCCGTG
	R	TGGACACCATACTTGAGCAGA
FABP4	F	TCACCATCCGGTCAGAGAGTA
	R	TGTCGTCTGCGGTGATTTCAT
c/EBPα	F	GGGAGAACTCTAACTCCCCCA
	R	GGAGGTGACTGCTCATCGG
ACC	F	TGGACCTAGAAGAGAAGGAGGG
	R	GCCAGAGATCCCCAAATCAGA

F and R are forward and reverse primers, respectively.

## Data Availability

The data used to support the findings of this study can be made available by the corresponding author upon request.

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
