# Peer review of "Yeast-Hydrolysate-Derived 1-Methyl-1,2,3,4-tetrahydro-β-carboline-3-carboxylic Acid Inhibits Fat Accumulation during Adipocyte Differentiation"

_foods, 2023, doi:10.3390/foods12183466_

Round 1

Reviewer 1 Report

The manuscript is interesting and might be useful to the nutritionist. however, the following comments and clarifications may imrove the manuscript quality and useful to the reader.

The author should mention hydrolysate derived from which yeast

Complete this sentence "This study aimed to identify the effect of yeast hydrolysate (YH) on lipogenesis, its action mechanism, and its active compounds." on what?

Line No. 23: 2 mg/mL is optimized value or on what basis author elected this value.

reduced the mRNA level of G6PD and malic enzyme. If possible mention the quantity of them.

improve the conclusion section of abstract

The author may provide Obesity data in authors country and global status

The background information about the study in introduction section need to be improved

Saccharomyces cerevisiae is only the yeast hydrolysate reported yet? listout anyother yeasts 

Nolvelty statement need to add and explain how this manuscript is fill the research gap in this field

The author may provide brief details about yeat hydrolysate extraction or source

Figure 1 microscopical images are not clear

Results interpration need to be scientifically improved

The author may use the recent references for the discussion section, since the author used very old references. There are more number of articles available, so author can refer them for discussion. 

In the entire manuscript, used references are too old

There are some typo and grammatical errors in the manuscript. The authors need to proofread the manuscript.

The manuscript need minor revision

Author Response

Thank you for your kind review of our manuscript.

please see uploaded response 

Reviewer 2 Report

Comments and Suggestions for Authors

This manuscript tests the effect and mechanism of yeast hydrolyzate (YH) on lipogenesis. The active compounds in YH with anti-adipogenesis effects were further confirmed through separation, purification and identification technology. The overall experimental purpose is clear, the methodology is appropriate, the data quality is good, and the results are adequately explained. Although there are some minor errors, they do not detract from the apparent contribution of this manuscript. This manuscript message is ideal for publication.

1. There are improper citations for some sources. Line 80-82, ref 18; Line 82-83, ref 21 (irrelevant for less than 10 kDa).

2. The image quality is extremely blurry; high-resolution images are required. Figure 1, Figure 4.

3. Figure 3 is repeated in Figure 4.

4. Line 222-224. The meaning of the sentence describing the 19.7-gram sample is unclear.

5. "data not shown" should be added. Line224-22; Line228-230.

6. In Materials and Methods, the manuscript should list the sources and specifications of YH acquisition.

Author Response

Comments and Suggestions for Authors
This manuscript tests the effect and mechanism of yeast hydrolyzate (YH) on lipogenesis. The active compounds in YH with anti-adipogenesis effects were further confirmed through separation, purification and identification technology. The overall experimental purpose is clear, the methodology is appropriate, the data quality is good, and the results are adequately explained. Although there are some minor errors, they do not detract from the apparent contribution of this manuscript. This manuscript message is ideal for publication.
1. There are improper citations for some sources. Line 80-82, ref 18; Line 82-83, ref 21 (irrelevant for less than 10 kDa).
<Response> Thank you for your comment. References are revised with proper citation.
Ref 18 in line 82 was revised to Ref 13, Ref 21 was removed.
2. The image quality is extremely blurry; high-resolution images are required. Figure 1, Figure 4.
<Response> We changed the figures with high-resolution images.
3. Figure 3 is repeated in Figure 4.
<Response> Figure 3 from Figure 4 was removed.
4. Line 222-224. The meaning of the sentence describing the 19.7-gram sample is unclear.
<Response> The indicated sentences have been revised like the following. “YH was subjected to solvent fractionation, and butanol fraction (19.7 g) was obtained through successive treatment with ethyl acetate and butanol. 9.4 g of butanol fraction was further subjected to RP-C18 silica gel chromatography, resulting in the separation into eight fractions” in line 257-259
5. "data not shown" should be added. Line224-22; Line228-230.
<Response> “data not shown” was added. In line 264, 266, 268.
6. In Materials and Methods, the manuscript should list the sources and specifications of YH acquisition.
<Response> The preparation and sources of YH were briefly described. A full explanation of YH preparation was limited due to the know-how of the company. “2.1 Preparation of yeast hydrolysate Yeast hydrolysate was prepared with slight modifications to the previous method. Saccharomyces cerevisiae cultured by Choheung Chemical was treated with proteolytic enzymes, and yeast hydrolysates of 10 kDa or less were recovered using a 10 kDa molecular weight cut-off membrane (Satocon cassette, Sartorius, Germany) and then spray dried. The total nitrogen content of YH was 650 mg/g and the active substance, MTCA, was 0.52 mg/g” in line 112-117.

Reviewer 3 Report

Review comments

The manuscript mainly discusses the inhibition of fat accumulation by yeast hydrolysate-derived 1-methyl-1,2,3,4-tetrahydro-β-carbo- 2 line-3-carboxylic acid.  All the gene names should be italicized.

1.   Abstract – include values.

2.   Introduction

a.    Even antiglycation end-product accumulation can lead to obesity. Highlight this point.

b.   Highlight the relationship between leptin and obesity.

c.    What was the negative control used in adipocyte differentiation assay?

d.   What was the housekeeping gene used?

3.   Materials and methods

a.      2.1- It is not clear what the authors measured. It should be clearly specified.

b.     Expand abbreviations when used first.

4.   Results

a.      Figure 1. The middle picture is not clear at all. It should be replaced.

b.     Data visualization should be improved.

5.   Discussion

a.   What is the main reason for the inhibition of adipocyte differentiation (see. Biomolecules 2020, 10, 165; doi:10.3390/biom10020165).

b.   It is not clear whether the authors further conducted studies using the isolated compound.

c.   What are the study limitations?

d.   How effective is this in controlling obesity?

Author Response

Thank you for your kind review comments on our manuscript

Please see uploaded our response 
